# Characterization of Divergent Grapevine Badnavirus 1 Isolates Found on Different Fig Species (*Ficus* spp.)

**DOI:** 10.3390/plants11192532

**Published:** 2022-09-27

**Authors:** Sergei Chirkov, Anna Sheveleva, Svetlana Tsygankova, Fedor Sharko, Irina Mitrofanova

**Affiliations:** 1Department of Virology, Faculty of Biology, Lomonosov Moscow State University, 119234 Moscow, Russia; 2National Research Center “Kurchatov Institute”, 123182 Moscow, Russia; 3Tsitsin Main Botanical Garden of Russian Academy of Sciences, 127276 Moscow, Russia

**Keywords:** *Ficus* spp., badnavirus, grapevine badnavirus 1, high-throughput sequencing

## Abstract

Fig mosaic disease is spread worldwide and is believed to have a viral etiology. Divergent isolates of grapevine badnavirus 1 (GBV1), named fGBV1, were discovered on *Ficus carica*, *F. palmata*, *F. virgata*, and *F. afghanistanica* in the fig germplasm collection of the Nikita Botanical Gardens, Russia, expanding the list of viruses infecting this crop. The complete genomes of five fGBV1 isolates from *F. carica* and *F. palmata* trees were determined using high-throughput and Sanger sequencing. The genomes comprised 7283 base pairs, contained four overlapping open reading frames, were 99.7 to 99.9% identical to each other, and related to GBV1 (83.2% identity). The reverse transcriptase RNase H genome regions of fGBV1 and GBV1 share 84.6% identity, indicating that fGBV1 is a divergent isolate of GBV1, which was found on the new natural hosts from a different family (*Moraceae*). Further, fGBV1-specific primers were developed to detect the virus using RT-PCR. Survey of 47 trees, belonging to four fig species and 14 local and introduced *F. carica* cultivars, showed the high fGBV1 prevalence in the collection (93.6%), including trees with no obvious symptoms of fig mosaic disease.

## 1. Introduction

Fig (*Ficus carica* L., *Moraceae*) is one of the most ancient fruit crops [1]. Originating in Asia Minor, fig has spread throughout the Near- and Middle East, the Mediterranean, and currently around the world, mainly in subtropical areas. The most serious disease of this crop is fig mosaic disease (FMD), common in many regions of its cultivation. The symptoms of the disease are diverse and are typically manifested on leaves and fruits as mottling, mosaics, ring spots, discoloration, or deformation [2,3]. It is believed that the disease has a viral etiology. Fifteen viruses from different taxonomic groups and three viroids were detected on fig. The symptoms of FMD are mainly attributed to fig mosaic virus (FMV, genus *Emaravirus*, family *Fimoviridae*), and their diversity is due to the influence of other viruses in mixed infection [4,5,6]. Among them, fig badnavirus 1 (FBV1, genus *Badnavirus*) was identified [7] and was the only representative of the family *Caulimoviridae* found on figs until recently.

Badnaviruses are widely distributed on fruit and ornamental plants. Their genomes are represented by a single molecule of non-covalently closed circular double-stranded DNA of 7–9 kbp. The genome is transcribed to produce a greater-than-genome length terminally redundant pregenomic (pg) RNA, which is either translated or serves as a template for replication of the viral genome through reverse transcription. The genome usually contains three or four open reading frames (ORF). The large intergenic region (LIGR), which is enclosed between the end of ORF3 or ORF4 and the beginning of ORF1, contains transcription regulatory motifs. By convention, the beginning of the genome is considered to be the first nucleotide of the tRNA^met^ binding site, which serves as a primer for the DNA minus strand synthesis. Badnaviruses are naturally transmitted by mealybugs and aphids in a semi-persistent manner [8,9,10]. 

The only fig germplasm collection in Russia is maintained in the Nikita Botanical Gardens (NBG), Yalta. It includes 267 local and introduced cultivars and forms of *F. carica* and trees of other fig species: *F. palmate* Forssk., *F. virgate* Roxb., and *F. johannis* subsp. *afghanistanica* (Warb.) Browicz) [11]. FMD symptoms were observed on about a third of the *F. carica* trees [12]. Metagenomic analysis of five symptomatic *F. carica* and *F. palmata* leaf samples revealed a substantial number of reads related to FMV, fig cryptic virus (FCV), and FBV1, as well as to one more badnavirus, which has not been identified on figs before. The Russian isolates of FMV and FCV were characterized previously [13,14]. 

The objectives of this work were assembly and characterization of the new fig badnavirus genome and study on the prevalence of this virus in the fig collection using RT-PCR with primers developed for its specific detection. This virus was shown to be a divergent variant of grapevine badnavirus 1 (GBV1, MF781082), recently discovered on grapevine in Croatia [15], and widespread in the collection, including trees with no obvious FMD symptoms. 

## 2. Results and Discussion

Total RNAs from the *F. carica* cultivars Temri, Kraps di Hersh, Bleuet, Smena, and from an *F. palmata* tree, displaying typical FMD symptoms [13], were used for metagenomic sequencing. On average, 695,000 quality-filtered 150-bp pair-ended reads per library were generated by high-throughput sequencing (HTS). The assembled contigs were aligned to the nucleotide (nt) sequences of badnaviruses (taxid:10652) available in GenBank. In total, BLASTn search found in five samples 12 contigs ranging from 122 to 7188 nt, which were 81.0–84.8% identical to the full-length genome of GBV1 [15] (query coverage 94–100%). At the same time, these contigs were 66.9–72.5% identical to the FBV1 complete genome sequences and thus seem to be derived from another badnavirus, which was designated fGBV1 for convenience and to discriminate the GBV1 isolates from grape and fig.

The complete genomes of five fGBV1 isolates were assembled from the contigs. The isolates from the *F. carica* cultivars and the *F. palmata* tree were 99.7–99.9% identical to each other, indicating the low level of genetic diversity of the virus and the high reliability of the HTS results as well. Joints and gaps between the contigs and the near entire LIGR were re-sequenced and fully confirmed by the Sanger method using custom primers flanking the regions of question (Appendix A). In addition, the Sanger sequencing of the LIGR confirmed the circular nature of the fGBV1 genome. 

The fGBV1 genome of the isolate Tem64 from the cultivar Temri comprises 7283 bp and includes four overlapping ORFs. The genome starts with the tRNA^met^ binding site (TGGTATCAGATAGTTT, positions 1–18). ORF1 (positions 287–718) and ORF2 (715–1122) encode the proteins 143 and 135 amino acid (aa) residues, respectively. ORF3 (positions 1119–6719) encodes a putative polyprotein 1866 aa length. Using the CD Search Service, zinc finger (aa 810–827), peptidase A3 (aa 1085–1285), reverse transcriptase (RT, aa 1311–1497), and RNase H (aa 1593–1721) motifs were identified in the polyprotein. The additional ORF4 is located at positions 6488–6748, overlapping the 3′-end of ORF3. The LIGR encompasses a genome segment at positions 6749 to 286 and comprises 821 nt. A TATA-box (TATTTAA, positions 7107–7113) was similar to that of commelina yellow mottle virus (X52938) [16], a species type of the genus *Badnavirus*. A putative polyadenylation signal (AATAAA) was identified at positions 7224–7229. In the remaining four isolates of the virus, all the functional elements of the genome mentioned above were at the same positions. Thus, the organization of the fGBV1 genome is typical of badnaviruses. Twenty-eight nt substitutions were randomly dispersed along the five genomes. In the coding regions, most of them (19 out of 23) turned out to be non-synonymous. 

Twenty-four genome sequences of approved and tentative badnaviruses, available in GenBank, were selected for phylogenetic analysis. These included badnaviruses close to GBV1 [15] and a number of virus species from more distant groups [9]. Since the five fGBV1 genomes were almost identical, only the isolate Tem64 was employed for phylogeny. Four phylogenetic trees based on the full-length genomes, ORF3 nt and aa sequences, and aa sequences of the RT/RNaseH domains (410 aa residues) were reconstructed.

When analyzing complete genomes, fGBV1 was shown to cluster with GBV1 (Figure 1). The sister clade was formed by FBV1 (JF411989) and grapevine Roditis leaf discoloration-associated virus (GRLDaV) (HG940503) [17]. Both clades and the whole cluster, which included these four viruses, were supported by the 100% bootstrap values. The composition of this phylogroup and its position on the three other trees were similar (Appendix A). The results of phylogenetic analysis also show that FBV1, GRLDaV, GBV1, and fGBV1 had a common ancestor, suggesting the possibility of vector transmission of GBV1 isolates from fig to grapes or vice versa ref. [17] (this work). This assumption requires experimental verification. 

Based on the phylogeny and complete genome identity data, fGBV1 is most closely related to GBV1. The close comparison of their genomes is presented in Table 1. It should be stressed that the direct comparison of the fGBV1 and GBV1 genomes was difficult because the latter was deposited in GenBank in a form, not quite common for badnaviruses. To compare, the GBV1 genome was presented here in the traditional manner, i.e., starting from the tRNA^met^ sequence.

Due to the high similarity of the five fGBV1 genomes, only one isolate (Tem64) was taken to compare. The fGBV1 genome is 138 nt longer mainly due to larger sizes of the LIGR and ORF3. In contrast, fGBV1 ORF4 is noticeably shorter because its start AUG codon is much closer to the LIGR than in GBV1. RT/RNaseH genome regions of these two viruses shared 84.6% identity at the nt level. According to the demarcation criteria of the genus *Badnavirus*, the differences between species in this region should be more than 20% [8,9]. Thus, fGBV1 should be regarded as a divergent isolate of GBV1. The differences between fGBV1 and GBV1 can possibly be due to their adaptation to different hosts. 

According to the ICTV, there are three criteria for assigning a certain badnavirus into a new species: host ranges, differences in polymerase (RT+ RNase H) nt sequences of more than 20%, and vector specificities [18]. GBV1 was found on grapevine (*Vitis vinifera*, *Vitaceae*), while fGBV1 was revealed on figs (*Ficus* spp., family *Moraceae*). Detection of fGBV1 on host plants from the different family could potentially be a reason to consider it a new virus species. However, although most badnaviruses have narrow natural host ranges, there are a few exceptions. For example, banana (family *Musaceae*) is a common host for banana streak viruses. At the same time, the banana streak CA virus isolate Hainan (OL803889) was detected on sugarcane from the family *Poaceae*. Canna yellow mottle virus was first detected and ubiquitous in cannes (*Cannaceae*). On the other hand, this virus was also found on betel nut (*Piper betel*, *Palmaceae*; e.g., KJ825824, KM373210, KM403570), ornamental ginger (*Alpinia purpurata*, *Zingiberaceae*; KU168312), and carnation (*Dianthus* spp., *Cariofillaceae*; KP836342). Thus, the second criterion also does not allow fGBV1 to be considered a distinct species.

The prevalence of fGBV1 in the collection was surveyed using RT-PCR. NucleoSpin RNA Plant Kit was chosen to isolate total RNA because the standard protocol involves on-column processing of the sample with recombinant DNase supplied with the kit. As primers for the GBV1 detection [15] did not recognize the fig isolates of the virus (data not shown), the fGBV1-F1/R1 primers were designed for the specific fGBV1 detection.

During a small-scale survey, amplicons of the expected size 633 bp were generated in each of 31 *F. carica* samples, including local and introduced cultivars, as well as in five *F. palmata*, one *F. afghanistanica*, and in seven of ten *F. virgata* specimens (Table 2). In total, 93.6% of the trees tested positive for fGBV1, indicating that it is widespread on *F. carica* and other fig species in the NBG fig germplasm collection.

Although the vast majority of the tested plants were shown to be fGBV1-infected, the virus was not detected in three *F. virgata* trees (Table 2, Figure 2, upper gel). These plants can be considered as negative controls, indicating the specificity of the PCR assay. Another badnavirus, FBV1, is also known to be widespread on *F. carica* worldwide [4,7]. Using RT-PCR with 1094F/1567R primers [7], FBV1 was shown to be widely distributed in the NBG fig collection. Some representative results of the FBV1 testing are demonstrated in Figure 2 (lower gel). Comparison of the upper and lower gels showed that fGBV1-specific primers did not recognize FBV1 in the *F. virgata* tree II/2/68, further supporting the specificity of the fGBV1 analysis. In addition, eight 633 bp amplicons were sequenced (Table 2). All these were fGBV1, confirming the virus-specific detection. Thus, the fGBV1-F1/R1 primers can apparently be used for specific detection of the virus.

Badnavirus nucleic acid can be in episomal and/or integrated forms in infected plant [9]. DNase-treated total RNA was used to detect fGBV1 by RT-PCR, resulting in the specific PCR product generation (Figure 3, lane 1). No amplification was observed when the RT step was omitted, indicating that DNA contaminations were efficiently removed (Figure 3, lane 2). Since fGBV1 was detected after the DNase treatment, viral pgRNA seemed to be the template for RT-PCR, suggesting the active virus replication at least in fig plants studied in this work. On the other hand, if the DNase treatment was omitted, a strong PCR product of the expected size was obtained by direct PCR (Figure 3, lane 3). This result shows that total RNA contained residual DNA in an amount sufficient to detect the virus by direct PCR. Thus, both PCR and RT-PCR assays can be used to detect fGBV1 successfully. Whether the virus DNA is also integrated in the fig genome has yet to be studied.

The sequences of the 633 bp PCR products were shown to be 99.8 to 100% identical at the nt and aa levels, irrespective of the fig species, *F. carica* cultivar, or the terrace on which the tested tree grows. Meanwhile, the five fGBV1 full-length genomes were also almost identical; all the tested fig trees were apparently infected with the same fGBV1 isolate, which could be disseminated along the collection by some invertebrate vector.

Most fGBV1 isolates were found on trees with typical FMD symptoms, which could be induced by either fGBV1 or other viruses (Table 2). However, fGBV1 was also detected in several samples with no obvious symptoms of the disease, including two trees of the local *F. carica* cultivar Sabrutsiya Rosovaya, as well as some *F. afghanistanica*, *F. virgata,* and *F. palmata* trees. This suggests that fGBV1 does not appear to cause symptoms on infected plants by itself or these trees are in the early stage of infection when the viral titer is still low. The observed FMD symptoms were likely due to FMV, which was detected in all the symptomatic trees listed in Table 2 using RT-PCR as described previously [13].

Thus, metagenomic analysis revealed a previously uncharacterized fig badnavirus that expands the list of viruses infecting this crop. Indeed, fGBV1 is a divergent isolate of GBV1 from grapevine and is clearly different from FBV1. Nevertheless, two fig badnaviruses, FBV1 [7] and fGBV1 (this work), have much in common. Both viruses are widely distributed in surveyed plantings. FBV1 was detected in the vast majority of fig samples of different origins [7] and is currently the most widespread fig virus in the world [4]. Further, fGBV1 was revealed in 93.6% of the samples tested in Russia (Table 2). Genomes of known FBV1 isolates are 99–100% identical [7]; likewise, genomes of the Russian fGBV1 isolates are just as conserved. Moreover, both viruses seem to be symptomless in figs. However, more fGBV1 isolates from other geographical regions should be studied to better understand the genetic diversity and incidence of the virus. Since fGBV1 was shown to be widespread in introduced cultivars, it is very likely that this virus can be detected in other regions of fig cultivation using the RT-PCR assay elaborated in the present study.

## 3. Materials and Methods

Plant material was gathered in the fig germplasm collection of the NBG (44.51N; 34.23E) from 2018 to 2020. The collection is located on a terraced slope and occupies three terraces elongated in the latitudinal direction. Self-rooted trees of various fig species and *F. carica* cultivars are randomly distributed within rows. Most trees tested for the viruses were about 30 years old. Leaf samples from each tree to be analyzed were tested individually.

Total RNA for HTS was extracted from the *F. carica* cultivars Temri, Kraps di Hersh, Bleuet, Smena, and from an *F. palmata* tree with typical FMD symptoms using RNeasy Plant Mini Kit (Qiagen, Hilden, Germany) according to the manufacturer’s protocol. DNA libraries were prepared using TruSeq Stranded Total RNA Library Prep Plant kit (Illumina, San Diego, CA, USA) and sequenced on MiSeq Illumina platform as described previously [13,14]. The raw sequence data are available at SRA in NCBI: PRJNA868530. Contigs were assembled de novo using metaSpades program version 3.14 [19]. Badnavirus-related contigs were identified using BLASTn (https://blast.ncbi.nlm.nih.gov/Blast.cgi) (accessed on 17 March 2022). The full-length genomes of five GBFIV isolates were deposited in GenBank under accession numbers OP087314 to OP087318.

ORFs in the complete virus genomes were identified using ORF Finder (https://ncbi.nlm.nih.gov/orffinder) (accessed on 17 March 2022). Conserved domains in the polyproteins were mapped using the Conserved Domain Search Service (https://ncbi.nlm.nih.gov/Structure/cdd/wrpsb.cgi) (accessed on 17 March 2022) (CDD v.3.20-59693 PSSMs). Multiple alignments of nt and deduced aa sequences using ClustalW, calculation of sequence identities, and phylogenetic analysis were performed in MEGA7 [20]. Phylogenetic trees were reconstructed using the neighbor-joining method and the Kimura 2-parameter or p-distance models (for nt and aa sequences, respectively).

The prevalence of fGBV1 in the collection was surveyed using RT-PCR. For this purpose, total RNA was extracted using NucleoSpin RNA Plant Kit (Macherey-Nagel, Dueren, Germany) according to the manufacturer’s instructions. Random hexamer primers and Moloney murine leukemia virus (MMLV) reverse transcriptase (both from Evrogen, Moscow, Russia) were used for the first-strand cDNA synthesis. Based on the virus sequences obtained by HTS, the fGBV1-F1 (5′-AGCGGCACGAAGGATAGCTA-3′) and -R1 (5′-TCCAGGTGATCTGTAACATGTT-3′) primers were developed for the specific fGBV1 detection. These primers target ORF3 at genome positions 1253–1272 and 1864–1885, respectively, amplifying a PCR product of 633 bp. The cycling conditions were 94 °C for 3 min, 35 cycles of 94 °C for 30 s, 50 °C for 30 s, 72 °C for 40 s, and a final extension at 72 °C for 7 min. PCR products were analyzed by 1.5% (*w*/*v*) agarose gel electrophoresis in TAE buffer and visualized by ethidium bromide staining. PCR products were purified from gel using BC022 Cleanup Standard kit (Evrogen) and sequenced in both directions by Evrogen. The resulting sequences (minus primers) were deposited in GenBank under accession numbers OP087319 to OP087326. FBV1 was tested by RT-PCR as described previously [7].

## Figures and Tables

**Figure 1 plants-11-02532-f001:**
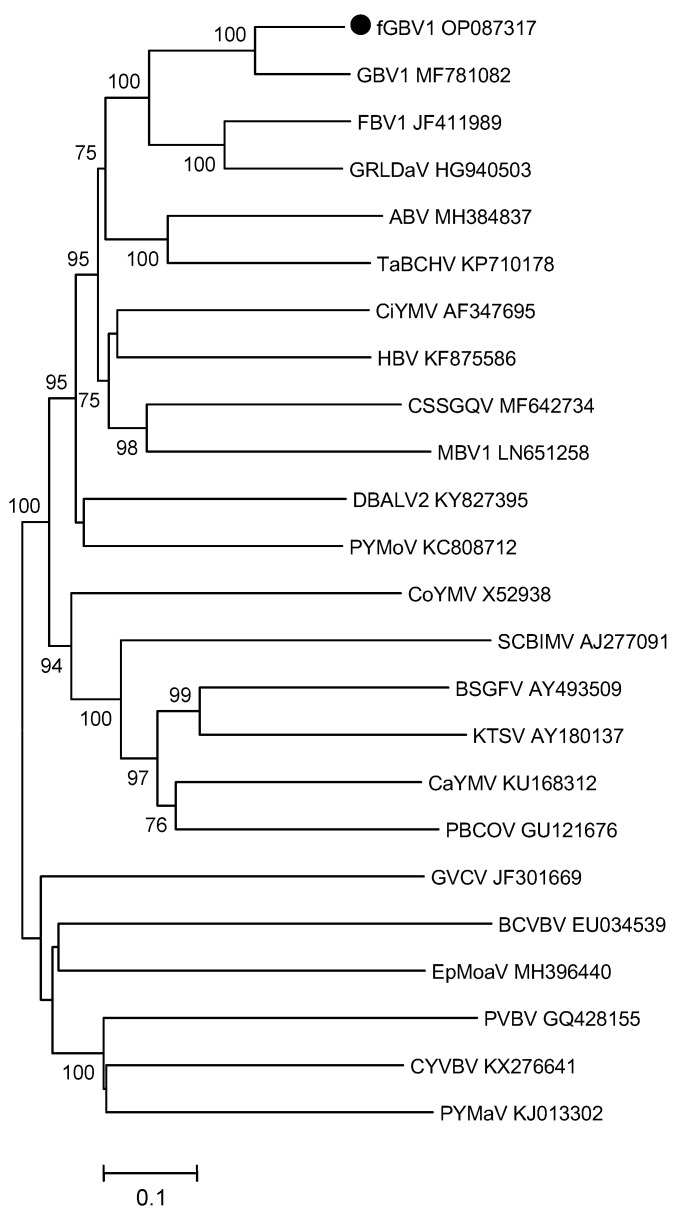
Phylogenetic analysis of complete genome sequences of members of the genus *Badnavirus*. The tree was reconstructed using the neighbor-joining algorithm implemented in MEGA7. Bootstrap values (from 1000 replicates) are indicated next to the corresponding nodes as percentage (>75%). The acronyms of virus names and accession numbers of isolates are shown at the end of branches. Abbreviated names of the viruses are as follows: ABV—Aglaonena bacilliform virus; BSGFV—Banana streak GF virus; BCVBV—Bougainvillea chlorotic vein banding virus; CSSGQV—Cacao swollen shoot Ghana Q virus; CYVBV—Cacao yellow vein banding virus; CaYMV—Canna yellow mottle virus; CoYMV—Commelina yellow mottle virus; DBALV2—Dioscorea bacilliform AL virus 2; EpMoaV—Epiphyllum mottle-associated virus; FBV1—Fig badnavirus 1; GBV1—Grapevine badnavirus 1; GRLDaV—Grapevive Roditis leaf discoloration-associated virus; GVCV—Grapevive vein clearing virus; HBV—Hibiscus bacilliform virus; KTSV—Kalanchoe top-spotting virus; MBV1—Mulberry badnavirus 1; PYMaV—Pagoda yellow mosaic associated virus; PVBV—Pelargonium vein banding virus; PBCOV—Pineapple bacilliform CO virus; PYMoV—Piper yellow mottle virus; SCBIMV—Sugarcane bacilliform IM virus; TaBCHV—Taro bacilliform CH virus. Grapevine badnavirus 1 from fig (fGBV1) is highlighted by a black circle (●).

**Figure 2 plants-11-02532-f002:**
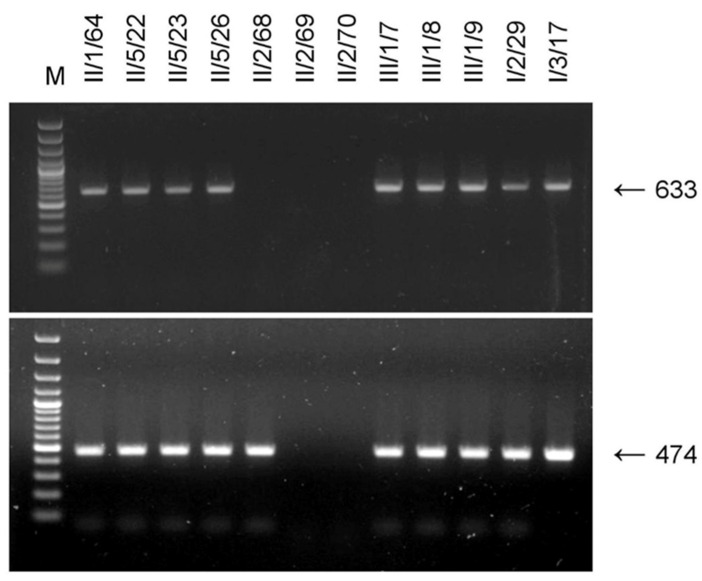
Agarose gel electrophoresis of PCR products generated by RT-PCR assay of fGBV1 (upper gel) and FBV1 (lower gel) in selected fig samples using fGBV1-F1/R1 (this work) and 1094F/1567R [7] primers, respectively. The tree numbers (see Table 2 for details) are shown above the picture. Arrows right of the pictures indicate PCR product of the corresponding size, bp. M—GeneRuler 100 bp DNA ladder Plus (Thermo Scientific).

**Figure 3 plants-11-02532-f003:**
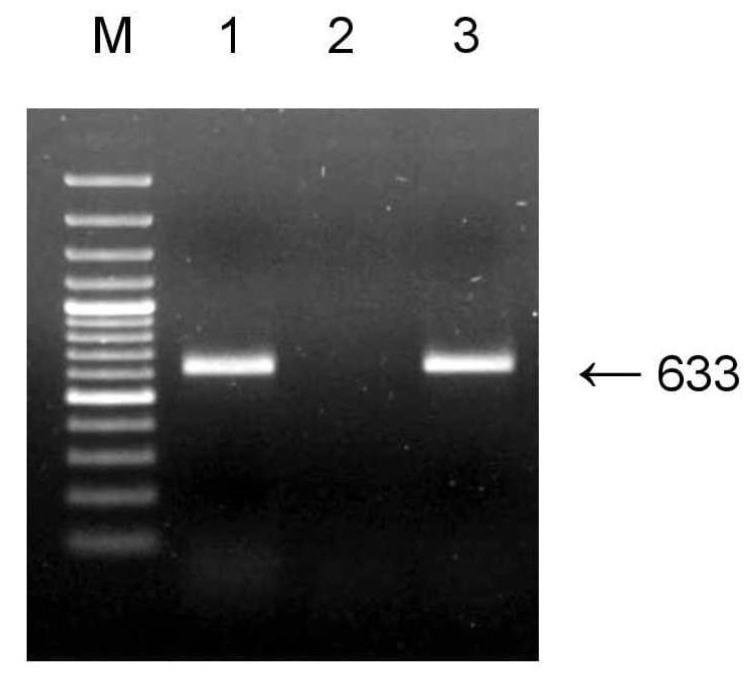
Effect of DNase treatment on fGBV1 isolate Tem64 detection by RT-PCR. Lane 1: conventional RT-PCR; lane 2: RT step omitted; lane 3: no DNase treatment. Arrows right of the pictures indicate PCR product of the corresponding size, bp. M—GeneRuler 100 bp DNA ladder Plus (Thermo Scientific).

**Table 1 plants-11-02532-t001:** Comparison of grapevine badna FI virus (GBFIV) and grapevine badnavirus 1 (GBV1) genomes ^a^.

Genome Region	nt Positions/Length (nt/aa) ^b^		Identity (nt/aa), %
	fGBV1	GBV1	
ORF1	287..718/(432/143)	273..704/(432/143)	85.2/85.3
ORF2	715..1122/(408/135)	701..1,114/(414/137)	84.6/90.4
ORF3	1119..6719/(5601/1866)	1111..6651/(5541/1846)	82.8/90.2
RT/RnaseH ^c^	5052..6281/(1230/410)	4696..5925/(1230/410)	84.6/95.6
ORF4	6488..6748/(261/86)	6240..6680/(441/146)	76.2/69.8
LIGR ^d^	6749..286/(821/-)	6681..272/(737/-)	87.3/-
Whole genome	1..7283	1..7145	83.2/-

^a^ Isolates Tem64 (fGBV1, OP087317) and VLJ-178 (GBV1, MF781082). ^b^ ORFs positions were determined using the Open Reading Frame Finder (https://ncbi.nlm.nih.gov/orffinder/) (accessed on 17 March 2022). ^c^ Positions of the RT/RNaseH motif were determined using the Conserved Domain Search Service (https://www.ncbi.nlm.nih.gov/Structure/cdd/wrpsb.cgi) (accessed on 17 March 2022). ^d^ Large intergenic region.

**Table 2 plants-11-02532-t002:** List of fig trees tested for grapevine badnavirus 1 (fGBV1) from fig ^a^.

Fig Species	Cultivar	Origin	Tree Location ^b^	Isolate	fGBV1 ^c^	FMD ^d^ Symptoms	Genomic Sequence	GenBank Accession Number
*Ficus* *carica*	Temri	Introduced	II/1/62	Tem62	+	+		
			II/1/63	Tem63	+	+		
			II/1/64	Tem64	+	+	Complete genome	OP087317
			II/1/65	Tem65	+	+		
			II/1/66	Tem66	+	+		
	Kraps di Hersh	Introduced	II/1/46	KDH46	+	+	ORF3 protein gene	OP087319
			II/1/48	KDH48	+	+	Complete genome	OP087315
			II/1/49	KDH49	+	+		
			II/1/50	KDH50	+	+		
	Belle Dure	Introduced	II/1/1	BD1	+	+		
			II/1/12	BD12	+	+	ORF3 protein gene	OP087320
			II/1/23	BD23	+	+		
			II/1/33	BD33	+	+		
			II/1/45	BD45	+	+		
	Bleuet	Introduced	I/3/17	Blu17	+	+	Complete genome	OP087316
	Figue Blanche	Introduced	I/3/7	FB7	+	+	ORF3 protein gene	OP087321
			I/3/8	FB8	+	+		
	Pomoriyskiy	Introduced	I/2/54	Pom54	+	+	ORF3 protein gene	OP087322
			I/2/56	Pom56	+	+		
	Ordubadskiy	Introduced	I/2/35	Ord35	+	+	ORF3 protein gene	OP087323
			I/2/36	Ord36	+	+		
	Die Dalmatie	Introduced	I/1/7	Dlm7	+	+		
	Smena	Local	II/4/17	SM17	+	+	Complete genome	OP087318
			II/4/20	SM20	+	+		
	Medovyiy	Local	I/3/37	Med37	+	+	ORF3 protein gene	OP087324
	Krymskiy Chernyiy	Local	I/3/75	KC75	+	+		
	Limonno- Zheltyiy	Local	I/4/40	LZ40	+	+		
	Violette	Local	I/3/84	Viol84	+	+		
	Sabrutsiya Rozovaya	Local	III/1/3	SR3	+	−		
			III/1/4	SR4	+	−		
			III/1/6	SR6	+	+		
*F. afghanistanica*	Not applicable	Introduced	I/2/29	Afg	+	−		
*F. virgata*	Not applicable	Introduced	II/5/2	Vir2	+	+		
			II/5/3	Vir3	+	+		
			II/5/22	Vir22	+	+		
			II/5/23	Vir23	+	+	ORF3 protein gene	OP087325
			II/5/24	Vir24	+	+		
			II/5/25	Vir25	+	−		
			II/5/26	Vir26	+	−		
			II/2/68		−	−		
			II/2/69		−	−		
			II/2/70		−	−		
*F. palmata*	Not applicable	Introduced	III/1/7	Pal7	+	+	ORF3 protein gene	OP087326
			III/1/8	Pal8	+	−		
			III/1/9	Pal9	+	−	Complete genome	OP087314
			III/1/11	Pal 11	+	+		
			III/1/13	Pal 13	+	+		

^a^ Using RT-PCR. ^b^ Number of terrace/number of row/number of tree in the row. ^c^ (+)—infected; (−)—not infected. ^d^ Fig mosaic disease. (+)—typical symptoms; (−)—no conspicuous symptoms.

## Data Availability

Sequencing data have been deposited in GenBank, and their accession numbers are provided within the article.

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
