# Peer review of "Characterization of Divergent Grapevine Badnavirus 1 Isolates Found on Different Fig Species (Ficus spp.)"

_plants, 2022, doi:10.3390/plants11192532_

Round 1

Reviewer 1 Report

This article describes interesting results on the distribution of a virus in different Ficus species but contrary to what is written in the title, it is not a new virus that is described but an isolate of the Grapevine Bacilliform Virus 1 species (GBV1)

1) Why is the point of creating a new virus name for a new isolate of GBV1, already recognized  species by ICTV? The title should be changed.

2) It is interesting to detect viral RNA to be sure replication is active and to avoid integrated DNA to be detected but DNA detection is probably more sensitive. It could have been interested to compare the sensitivity of the two methods.

3) Also in Mat & Methods

Which alignment algorithm has been used ?

The neighbor joining mehtod is not the best method for the construction of phylogenetic trees based on sequence alignments, the maximum likelihood is much better. To obtain a more robust tree, the method should be changed

Author Response

Reviewer 1.

This article describes interesting results on the distribution of a virus in different Ficus species but contrary to what is written in the title, it is not a new virus that is described but an isolate of the Grapevine Bacilliform Virus 1 species (GBV1)

1) Why is the point of creating a new virus name for a new isolate of GBV1, already recognized  species by ICTV? The title should be changed.

RE: The new title is: "Characterization of divergent grapevine badnavirus 1 isolates found on different fig species (Ficus spp.). Also, for convenience and to discriminate GBV1 isolates from grapevine and fig, the last ones were designated fGBV1 throughout the manuscript.

2) It is interesting to detect viral RNA to be sure replication is active and to avoid integrated DNA to be detected but DNA detection is probably more sensitive. It could have been interested to compare the sensitivity of the two methods.

RE: We treated total RNA isolated from infected plants with DNase followed by RT and PCR, obtaining an amplicon of the expected size. If the RT step was omitted, the virus-specific PCR product was not generated (lines 212-223, Fig. 3). This shows that viral RNA is present in the infected plant and hence the viral replication is active. Apparently, DNase treatment eliminates also plant DNA along with integrated viral DNA, if any. Conventional direct PCR and RT-PCR were approximately equal sensitive (data not shown in the paper), but qPCR was not performed. These studies may be the subject of further work.

3) Also in Mat & Methods

Which alignment algorithm has been used?

RE: ClustalW implemented in MEGA7. This information was added to the Material and Method section.

The neighbor joining mehtod is not the best method for the construction of phylogenetic trees based on sequence alignments, the maximum likelihood is much better. To obtain a more robust tree, the method should be changed

RE: Phylogenetic trees were reconstructed using the ML method. The topology of NJ and ML trees was the same, but some bootstrap values differed slightly (1-5% in both sides). Thus, in this case, both methods allowed to reconstruct equally robust trees. Therefore, we decided to leave the NJ method.

Reviewer 2 Report

The manuscript represents new information and genome analysis about the new badnavirus infecting fig plants.  

The paper is written; clearly, the experiments were set up correctly, and the right analysis was done on the data. 

Author Response

Thank you very much

Reviewer 3 Report

The article of Chirkov et al. entitled „ Characterization of a new fig (Ficus spp.) virus distantly related to grapevine badnavirus 1” reports the virus infection of fig germplasm collection in the Nikita Botanic garden in Russia. Two of the viruses present in these fig plants were reported earlier (fig mosaic virus and fig cryptic virus). In the present study, a new badnavirus is reported, which was not reported on this host previously. The virus was detected by metagenomics analysis, and RT/PCR assay was also developed for the detection of the prevalence of this virus in fig collection.

The paper is basically well written and the results are clearly presented.

 Generally, I have doubts about the description of the identified virus as a “new” bandavirus. According to the demarcation criteria of the genus Badnavirus, the difference between species in the RT/RNaseH region should be more than 20 %, but in the present study, it is just 15,4%. On the other hand, the proposed name “grapevine badna FI virus “contains the name of two plant species (grapevine and fig), even if there is no information if the identified virus is able to infect the grapevine or not. Generally, the survey of the host range is highly lacking in respect to describing a new virus. In ln 112-113 “the possibility of 112 vector transmission of these viruses from fig to grapes or vice versa” should be verified experimentally. In ln 147 the statement “Detection of GBFIV on host plants from the different families could potentially be a reason to consider it a new virus species” should be deleted, since no experimental data is available in this respect, and even in the next sentences several viruses are mentioned with different host plants.

Even it is described in lines 103-104 the construction of four phylogenetic trees (based on the full-length genomes, ORF3 nt and aa sequences, and aa sequences of the RT/RNaseH domains), but unfortunately, just the phylogenetic tree based on the full-length genome sequences is presented in the manuscript (Fig 1). It would be interesting to see the others, whether all of these phylogenetic trees have the same structure or not. If interesting differences could be detected in the location of the different viruses, recombination analysis could result in interesting observations.

The extension of Table 1 with the data of the viruses in the sister clade (FBV1 and GRLDaV) and also the discussion section related to this table (ln 137-144) would be very interesting in relation to the phylogenetic analysis.

It is obvious according to Fig 2 that “Comparison of the upper and lower gels showed that GBFIV-specific primers did not recognize FBV1 in the F. virgata tree” (ln 181-182). The question is, whether the opposite also true if the FBV1-specific primer did not recognize GBFIV or not.

Symptom induction is discussed in ln 234-240. Since the GBFIV was present both in plants with and without symptoms and the discussion suggests that the observed symptoms are typical in the case of FMV infection. So the detection of this FMV in the samples is essential, and the result of this analysis should be included in Table 2.

Generally, the results are interesting, but due to the sequence similarity of the RT/RNaseH genome region, I suggest changes in the concept of the presentation, just describing the identified virus on a new host, not as a new virus species.

Author Response

Reviever 3.

The article of Chirkov et al. entitled „ Characterization of a new fig (Ficus spp.) virus distantly related to grapevine badnavirus 1” reports the virus infection of fig germplasm collection in the Nikita Botanic garden in Russia. Two of the viruses present in these fig plants were reported earlier (fig mosaic virus and fig cryptic virus). In the present study, a new badnavirus is reported, which was not reported on this host previously. The virus was detected by metagenomics analysis, and RT/PCR assay was also developed for the detection of the prevalence of this virus in fig collection.

The paper is basically well written and the results are clearly presented.

 Generally, I have doubts about the description of the identified virus as a “new” bandavirus. According to the demarcation criteria of the genus Badnavirus, the difference between species in the RT/RNaseH region should be more than 20 %, but in the present study, it is just 15,4%. On the other hand, the proposed name “grapevine badna FI virus “contains the name of two plant species (grapevine and fig), even if there is no information if the identified virus is able to infect the grapevine or not.

RE: The virus we found is a divergent variant of grapevine badnavirus 1 (GBV1) detected on new hosts - different fig species, as was clearly stated in the manuscript. The word "new" means only that this badnavirus is new to figs. To avoid possible ambiguity, GBV1 from fig was designated in the revised MS as fGBV1, where "f" (fig) indicates the new host of the virus. See also the replies to Reviewer 1.

Generally, the survey of the host range is highly lacking in respect to describing a new virus. In ln 112-113 “the possibility of 112 vector transmission of these viruses from fig to grapes or vice versa” should be verified experimentally.  

RE: The results of our work show that GBV1 can naturally infect different fig species thus expanding the host range of the virus. We agree that the possibility of the fGBV1 transmitting from fig to grapevine needs experimental verification. This phrase was inserted to the manuscript. However, the study of the fGBV1 host range was out the scope of this work.

In ln 147 the statement “Detection of GBFIV on host plants from the different families could potentially be a reason to consider it a new virus species” should be deleted, since no experimental data is available in this respect, and even in the next sentences several viruses are mentioned with different host plants.

RE: According to the ICTV, there are three criteria for assigning a certain badnavirus into a new species: host ranges, differences in polymerase (RT+ RNase H) nt sequences of more than 20%, and vector specificities (see new reference [18] added to the text and the Reference section).  

[18] https://ictv.global/report/chapter/caulimoviridae/caulimoviridae/badnavirus

As GBV 1 was detected on new hosts, it should have been to test another possibility to assign the fig isolate to new badnavirus species (in addition to the level of the nt identity of the polymerase sequences). Therefore, we decided not to delete the sentence in line 147.

Even it is described in lines 103-104 the construction of four phylogenetic trees (based on the full-length genomes, ORF3 nt and aa sequences, and aa sequences of the RT/RNaseH domains), but unfortunately, just the phylogenetic tree based on the full-length genome sequences is presented in the manuscript (Fig 1). It would be interesting to see the others, whether all of these phylogenetic trees have the same structure or not. If interesting differences could be detected in the location of the different viruses, recombination analysis could result in interesting observations.

RE: The trees based on ORF3 nt and aa sequences, and aa sequences of the RT/RNaseH domains are presented in the supplementary material (Fig. S1A,B,C). All the four trees have the very similar topology. Recombination analysis of the badnavirus complete genome alignments using RDP4 detected no recombination in the GBV1, GBV1F, FBV1, and GRLDaV that make up this clade.

The extension of Table 1 with the data of the viruses in the sister clade (FBV1 and GRLDaV) and also the discussion section related to this table (ln 137-144) would be very interesting in relation to the phylogenetic analysis.

RE: Two divergent isolates of GBV1 were compared in Table 1. In contrast, FBV1 and GRLDaV are different viruses. We think that the addition of these viruses to Table 1 would be excessive and make it difficult to analyze the results presented in Table 1.

It is obvious according to Fig 2 that “Comparison of the upper and lower gels showed that GBFIV-specific primers did not recognize FBV1 in the F. virgata tree” (ln 181-182). The question is, whether the opposite also true if the FBV1-specific primer did not recognize GBFIV or not.

RE: We failed to find a GBV1F-infected and, simultaneously, FBV1-free tree so far to address this question. However, the FBV1-specific primers are not recognized the fGBV1 sequences in silico.

Symptom induction is discussed in ln 234-240. Since the GBFIV was present both in plants with and without symptoms and the discussion suggests that the observed symptoms are typical in the case of FMV infection. So the detection of this FMV in the samples is essential, and the result of this analysis should be included in Table 2.  

All the plants displayed FMD symptoms (Table 2) tested positive for FMV using RT-PCR as described in [13]. The corresponding phrase was rewritten as follows: "The observed FMD symptoms were likely due to FMV, which was detected in all the symptomatic trees listed in Table 2, using RT-PCR as described previously". Therefore, we decided not to make changes in Table 2.

Generally, the results are interesting, but due to the sequence similarity of the RT/RNaseH genome region, I suggest changes in the concept of the presentation, just describing the identified virus on a new host, not as a new virus species.

RE: The concept of the presentation was changed. The another badnavirus found on fig was characterized as a divergent variant of GBV1, not a new badnavirus.  

Reviewer 4 Report

I have no particular comments about the manuscript which seems to me to respond clearly to Plants guidelines. From the experimental point of view, the high-throughput sequencing approach is  widely used for the search for new viral agents, here it was applied correctly and the results are well presented and discussed. To my opinion before its publication, authors should clarify some elements or stylistic passages indicated below:

Title: why 'distantly' related? it seems clearly that we are facing a variant of the badnavirus species infecting the grapevine, in fact as authors discussed at line 143 'According to the demarcation criteria of the genus Badnavirus, the differences between species in this region should be more than 20%....GBFIV should be regarded as a divergent isolate of GBV1' and even at line 155 'the second criterion also does not allow GBFIV to be considered a distinct species'. finally, the phylogentetic tree clearly demonstrates that GBFIV is closely related to GBV1 rather than distantly. Then to my opinion, the fact that GBFIV was found for the first time in fig does not exclude that it may  be also present in other tree species of the same or other families.

Introduction: line 33, 'Apparently, FMD has the viral etiology', this sentence is cryptic and unclear to those who do not know the disease. It is necessary to explain it better, and frame the meaning well, also using the appropriate literature.

Table 2: I suggest a general revision of the column title:

'Origin' instead of 'Cultivar/species origin'

'Isolate' istead of 'Isolate name'

'GBFIV' instead of 'GBFIV detected'

'Genomic sequence' istead of Genome region sequenced

Line 235-238: given that there is no data relating to the vector preference, nor for what concerns the moment of infection, we cannot exclude that these are plants in the early stages of infection when the viral titer is still low. The method of detection based on endpoint PCR and not qPCR does not allow comparisons, it would have been interesting to use qPCR primers to compare the viral titer in the different Ficus species. Thus the absence of symptoms may also be related to a lower susceptibility of certain species/genotypes to infection. The authors should also consider these aspects in the discussion.

Author Response

Reviever 4.

I have no particular comments about the manuscript which seems to me to respond clearly to Plants guidelines. From the experimental point of view, the high-throughput sequencing approach is widely used for the search for new viral agents, here it was applied correctly and the results are well presented and discussed. To my opinion before its publication, authors should clarify some elements or stylistic passages indicated below:

Title: why 'distantly' related? it seems clearly that we are facing a variant of the badnavirus species infecting the grapevine, in fact as authors discussed at line 143 'According to the demarcation criteria of the genus Badnavirus, the differences between species in this region should be more than 20%....GBFIV should be regarded as a divergent isolate of GBV1' and even at line 155 'the second criterion also does not allow GBFIV to be considered a distinct species'. finally, the phylogentetic tree clearly demonstrates that GBFIV is closely related to GBV1 rather than distantly. Then to my opinion, the fact that GBFIV was found for the first time in fig does not exclude that it may be also present in other tree species of the same or other families.

RE: Borderline between closely and distantly related isolates of the same virus is often conditional. But in this case we are undoubtedly dealing with two divergent isolates of the same virus. Therefore, the word "distantly related" was replaced by "divergent" throughout the manuscript.

Introduction: line 33, 'Apparently, FMD has the viral etiology', this sentence is cryptic and unclear to those who do not know the disease. It is necessary to explain it better, and frame the meaning well, also using the appropriate literature.

RE: The phrase was rewritten as follows: "It is believed that the disease has a viral etiology. Fifteen viruses from different taxonomic groups and three viroids were detected on fig. The symptoms of FMD are mainly induced by fig mosaic virus (FMV, genus Emaravirus, family Fimoviridae), and their diversity is due to the influence of other viruses in mixed infection [4-6]".

Table 2: I suggest a general revision of the column title:

'Origin' instead of 'Cultivar/species origin'

'Isolate' istead of 'Isolate name'

'GBFIV' instead of 'GBFIV detected'

'Genomic sequence' istead of Genome region sequenced

RE: All the suggestions were corrected. The acronym "GBFIV" was replaced by "fGBV1".

Line 235-238: given that there is no data relating to the vector preference, nor for what concerns the moment of infection, we cannot exclude that these are plants in the early stages of infection when the viral titer is still low.

RE: The phrase was rewritten as follows: "This suggests that fGBV1 does not cause symptoms on infected plants by itself or these trees are in the early stage of infection when the viral titer is still low".

The method of detection based on endpoint PCR and not qPCR does not allow comparisons, it would have been interesting to use qPCR primers to compare the viral titer in the different Ficus species.

RE: We agree that such work should be done using qPCR. This subject can be investigated in further work.

Thus the absence of symptoms may also be related to a lower susceptibility of certain species/genotypes to infection. The authors should also consider these aspects in the discussion.

RE: This explanation is less likely, because of the three genetically identical trees of the cultivar Sabrutsiya Rozovaya (obtained by vegetative propagation), all were infected with fGBV1 but symptoms were found on only one. The similar situation is with F. palmata trees 7, 8, and 9 (Table 2).